# Introduction of Nanomaterials to Biosensors for Exosome Detection: Case Study for Cancer Analysis

**DOI:** 10.3390/bios12080648

**Published:** 2022-08-17

**Authors:** Myoungro Lee, Jinmyeong Kim, Moonbong Jang, Chulhwan Park, Jin-Ho Lee, Taek Lee

**Affiliations:** 1Department of Chemical Engineering, Kwangwoon University, Seoul 01897, Korea; 2School of Biomedical Convergence Engineering, Pusan National University, Yangsan 50612, Korea

**Keywords:** biosensors, cancer, exosomes, nanomaterials

## Abstract

Exosomes have been gaining attention for early cancer diagnosis owing to their biological functions in cells. Several studies have reported the relevance of exosomes in various diseases, including pancreatic cancer, retroperitoneal fibrosis, obesity, neurodegenerative diseases, and atherosclerosis. Particularly, exosomes are regarded as biomarkers for cancer diagnosis and can be detected in biofluids, such as saliva, urine, peritoneal fluid, and blood. Thus, exosomes are advantageous for cancer liquid biopsies as they overcome the current limitations of cancer tissue biopsies. Several studies have reported methods for exosome isolation, and analysis for cancer diagnosis. However, further clinical trials are still required to determine accurate exosome concentration quantification methods. Recently, various biosensors have been developed to detect exosomal biomarkers, including tumor-derived exosomes, nucleic acids, and proteins. Among these, the exact quantification of tumor-derived exosomes is a serious obstacle to the clinical use of liquid biopsies. Precise detection of exosome concentration is difficult because it requires clinical sample pretreatment. To solve this problem, the use of the nanobiohybrid material-based biosensor provides improved sensitivity and selectivity. The present review will discuss recent progress in exosome biosensors consisting of nanomaterials and biomaterial hybrids for electrochemical, electrical, and optical-based biosensors.

## 1. Introduction

Cancer is characterized by various complex symptoms depending on the type and site of occurrence [1,2]. Therefore, accurately predicting and diagnosing cancer can change the prognosis of patients considerably [3,4]. The current most effective way to diagnose cancer is to identify cancerous tissue and cancer cells through surgical biopsy. However, a surgical biopsy cannot be used to diagnose all types of cancer [5]. The presence of several cancerous tumors, such as breast cancer, can be confirmed via X-ray and ultrasound examination with expected surgical follow-up, whereas for nonspecific and deep organ cancers, such as pancreatic cancer, computed tomography and magnetic resonance imaging are required for diagnosis and staging. However, even these methods cannot easily differentiate between tumor and inflammation, and a surgical biopsy is difficult to perform [6,7]. Moreover, as cancer is not static, the loss of time and money during the biopsy process is a major limitation for the patient [8], and additional complications or side effects, such as cancer metastasis, may occur [9].

Recently, we have observed a paradigm shift towards liquid biopsy-based molecular analysis to supplement surgical biopsy [10,11]. A liquid biopsy is performed on the body fluid of a patient and facilitates the direct detection of cancer cells circulating in the body, tumor DNA, or tumor-derived material, with the advantages of real-time observation [12], noninvasiveness [13], and low cost [8,14]. This can reduce the burden of biopsy on the patient. The primary markers for liquid biopsy are the transmembrane receptor glycoprotein podoplanin [15,16], circulating tumor DNA [17,18], and exosomes [19,20].

Podoplanin is a transmembrane mucin-like glycoprotein expressed in various types of cancers and influences tumor cell migration and metastasis [15,21]. A circulating tumor cell chip composed of podoplanin antibody has been used to capture malignant pleural mesothelioma cells in preclinical models [22]. Blood levels of circulating tumor DNA are known to be higher in patients with cancer than in healthy individuals [23]. Analysis of circulating tumor DNA has improved clinical outcomes in some cancer types, such as nonsmall cell lung cancer [24] and breast cancer [25]. Extracellular vesicles (EVs) are cell-derived vesicles composed of a lipid bilayer, which retain protein and nucleic acid information of the parent cell, attracting attention as markers of cancer and disease [26,27,28]. EVs exist in a variety of sizes. Therefore, they can be divided into exosomes, small EVs with a size of 30–100 nm, and ectosomes, microvesicles, and apoptosis, which are large EVs with a diameter of 100–1000 nm or more. In general, small EVs are known to affect the transport of molecules related to cellular activity [29], while large EVs contain proteins and lipids involved in disease progression [30]. Therefore, it is an attractive material as a marker for liquid biopsy for diagnosing diseases. However, it has been reported that exosomes are more abundantly distributed compared to large oncosomes in studies targeting cancer cells [31]. Therefore, exosomes, which are small EVs, are evaluated as more valuable from an oncological point of view. In addition, from the viewpoint of early cancer diagnosis, cancer cell-derived exosomes are secreted at high concentrations in the early stages of tumor development [32]. Furthermore, exosomes are easy to identify, attracting attention as optimal biomarkers in liquid cancer biopsies. Recently, efforts have been made to detect exosomes based on various detection platforms, such as on electricity- [33,34,35,36,37,38,39], electrochemical- [40,41,42,43,44,45,46,47,48], and optical-based platforms [49,50,51,52,53,54,55,56].

However, in most research cases, the detection of exosomes is focused on the selective detection of specific cell-derived exosomes. From the perspective of liquid biopsy for cancer diagnosis, increasing the concentration of exosomes in the body suggests the possibility of developing cancer and related diseases. However, because exosomes in the body are secreted by various cells, they show slightly different differences in the composition, size, shape, and distribution of surface proteins [57]. Therefore, to diagnose cancer based on a transparent relationship with cancer, such as the case where melanoma-derived exosomes were detected in bodily fluid samples based on the CSPG4 protein expressed in melanoma, phenotypic issues need to be specified prior to experimentation. Recently, studies have been conducted to characterize the phenotype of exosomes using IR spectroscopy [58,59], optical methods [60,61,62], and cryo-transmission electron microscopy-based analysis methods [57].

In recent decades, various studies have been conducted to develop biosensors in the field of disease diagnosis, such as blood glucose sensors and pregnancy diagnostic kits. Biosensors focus on the rapid and quantitative detection of target molecules and can detect targets with high affinity based on biological receptors, such as enzymes, antibodies, and aptamers, which are designed nucleic acid sequences. A biosensor is largely composed of a detection unit, which is composed of a receptor, and a signal conversion unit. The signal generated from the detection unit is converted by the signal conversion unit into various signals, e.g., electrical and optical, thereby enabling quantification of the detected signal [63]. Since the discovery of the relationship between biomolecules and diseases, biosensors have been actively studied for the early diagnosis and treatment of diseases.

Existing enzyme-linked immunosorbent assays (ELISAs) and polymerase chain reaction (PCR) analyses can perform high-sensitivity analysis; however, they require expensive equipment and long analysis times. Moreover, the introduction of electrical and electrochemical analysis platforms has the advantage of being suitable for point-of-care diagnosis because it does not require labeling of the target molecule. However, previously reported biosensors have several limitations, such as low electrochemical signal strength, low stability, and low sensitivity to biomolecules [64].

To increase signal sensitivity, a highly sensitive bioprobe can be developed by increasing the surface area of the electrode or increasing the number of probes by increasing the surface roughness [65], which is aimed at increasing the activity, electrochemical electron transport, etc. Moreover, the introduction of nanomaterials to the electrode interface causes new nan-physical phenomena, such as plasmons, metal-enhanced fluorescence (MEF), and surface-enhanced Raman spectroscopy (SERS), leading to the development of a new type of biosensor [66,67]. Nanomaterials, such as noble metal nanoparticles, carbon nanoparticles [37], chalcogen compounds [33], and metal-organic frameworks [51], have excellent physical properties, leading to numerous developments in the manufacture of nanobiosensors.

Recently, the application of nanobiohybrid materials in manufacturing biosensors to overcome these problems has been discussed [68]. The application of conductive polymers and porous materials improves the electron transport reaction; based on the plasmonic and optical properties, the electrochemical signal increases, and the activity of biomolecules can be maintained for a long time [69,70]. In addition, by extending the sensor surface, more probes can be immobilized to improve the sensitivity [71]. Therefore, a biosensor to which nanobiomaterials are applied can be utilized to detect lower and wider exosomes based on their concentration in the body [72].

Although various studies for detecting exosomes have been conducted, a sensor that universally detects exosomes and can be commercialized has not been reported yet. This review aims to summarize the existing biosensor platforms for detecting exosomes. It will also suggest future directions for sensor research concerning exosome detection in liquid biopsies by identifying the limitations of current technologies. For the development of a biosensor composed of a hybrid of nanomaterials and biomaterials in precious metals, transition metals, carbon-based materials, and organometallic framework- (MOF) based materials will be introduced as key cases.

## 2. Pretreatment and Characterization of Cell-Derived Exosomes

### 2.1. Isolation of Exosomes in Samples

Extracellular vesicles, including exosomes, can be derived from various cells in the body [73]. In most sensor studies for exosome detection, exosomes derived from cancer cells cultured in laboratory conditions were evaluated [34,49,74], and in order to extend this to detection in clinical samples, exosomes were used before being applied to sensors. They need to be isolated and quantified from various biofluids such as blood, urine, and peritoneal fluids. There is a method of precipitating exosomes by ultracentrifugation of 100,000× *g* or more [75] for the isolation of exosomes and using common proteins (CD63, CD9, CD81) and hair cell-derived proteins present on the surface of exosomes as markers. Representative examples include an immune-based separation method for capturing exosomes in body fluids [76] and a size-based method for filtration using a membrane filter [77] considering that the diameter of general exosomes is 100 nm or less. In addition, studies have been conducted to isolate exosomes based on size-exclusion chromatography [78] and density gradient ultracentrifugation [79].

To isolate exosomes from mesenchymal stem cell cultures, we proposed a sucrose cushion ultracentrifugation method that improved the existing differential ultracentrifugation method [80] (Figure 1A). The existing ultracentrifugation method is one of the most representative and used methods for exosome separation, but several steps of centrifugation are required for precise separation. In addition, proteins in cells and cultures can interfere, and an increase in centrifugation steps results in a loss of exosome yield. The culture medium was centrifuged stepwise at 300× *g* and 10,000× *g* to remove cell debris and microvesicles in the proposed experimental method. Then, the medium was loaded in a 30% sucrose cushion solution similar to the exosome density, and centrifugation was performed under 100,000× *g* conditions. The proposed method recovered approximately 1.5–2 times more exosomes than the conventional method, while maintaining ultracentrifugation as the first step.

Sharma et al. proposed an immunoaffinity isolation method to capture exosomes in body fluids of melanoma patients [81] (Figure 1B). Based on the CSPG4 protein specifically expressed in melanoma, mAb 763.74 that binds to it was introduced to detect cancer-derived exosomes selectively. The antibody was first biotinylated and incubated with the exosome sample, and the biotinylated antibody-labeled exosome was captured through a column filled with streptavidin beads. For the recovery of unbound exosomes, the corresponding process was performed iteratively. The proposed isolation method was able to selectively capture melanoma-derived exosomes from venous blood samples in melanoma patients. The parent cell-specific protein-based immunoaffinity assay will be able to purify target cell-derived exosomes from various exosome samples.

### 2.2. Analysis of the Exosome Phenotype

Extracellular vesicles, such as exosomes, exhibit diversity in size, shape, protein composition, density, and distribution. Studies to characterize exosomes based on these characteristics have continued, but standards for clearly distinguishing them have not yet been established. Furthermore, because the classification of exosomes with submicro sizes by laboratory methods is limited [82], in general, dynamic light scattering (DLS), nanoparticle tracking analysis (NTA), and scattering flow cytometry methods used for exosome analysis can only check the physical information of exosomes [83]. In addition, the problem of exosome contamination is pointed out in the fluorescence-based analysis method. Finally, immunoblotting and western blotting methods can identify the constituent proteins of exosomes, but cost and time are limiting factors in identifying all proteins. Recently, an analysis method using IR spectroscopy, optical analysis with a specific label, and cryo-transmission electron microscopy has been reported. The characterization of these exosomes is meaningful as a factor for confirming the physical properties of exosomes according to diseases.

Yliperttula et al. proposed the characterization of extracellular vesicles based on infrared and Raman spectroscopy [59]. Evaluation of the particle-to-protein ratio based on NTA and bicinchoninic acid assay (Pa/Pr) for EVs purified by different purification methods from two different cells: identification of protein using western blotting and lipid-to-lipid ratio using ATR-FTIR. Unnecessary protein removal was performed through protein ratio measurements (Li/Pr) and Raman spectroscopy-based spiking. In the proposed study, it was possible to compare the number of exosome particles according to the purification method based on comparing Pa/Pr and protein markers. In addition, the purity of exosomes was evaluated through spectral comparison of ATR-FTIR and Raman spectroscopy, and ATR-FTIR showed an error rate of 6.61–10.26% in the blind test compared with NTA.

Varga et al. proposed a method for quantifying extracellular vesicles in biofluids using Flu-SEC and microfluidic resistive pulse sensing combined with size exclusion chromatography and a specific fluorescent label [60]. Experiments were conducted using a fluorescent antibody (PE-antiCD235a) that specifically binds to the red blood cell membrane protein glycophorin A, and wheat germ agglutinin markers bound with Alexa647 dye were analyzed for particle size distribution using MRPS and lipids using FTIR spectroscopy. Additionally, protein composition was analyzed. Finally, fluorescence signal analysis according to the binding of the red blood cell-specific antibody to the two concentrations of red blood cell-derived EV was performed through Flu-sec, and it was possible to determine that the sample EV was red blood cell-derived EV. The proposed system has the advantage of being able to quantify EVs without using equipment, such as HPLC.

Based on direct stochastic optical reconstruction microscopy (dSTORM), we proposed a method for visualizing extracellular vesicles and analyzing the domains present on the EV surface [62]. Using cells expressing CD63 with a green fluorescent protein tag and CD81 with a mCherry tag, we observed whether cell-derived EVs expressed the fluorescent tag protein of parental cells through dSTORM. EVs derived from parental cells with fluorescent protein expression factors expressed fluorescently tagged tetraspanin domains, and it was confirmed that the fluorescently tagged CD81 antibody binds to the EV surface even in wild-type EVs. The existence of microdomains on the EV surface was cross verified through cryogenic electron microscopy, and dSTORM visualized EVs in 3D with high resolution while overcoming the problems of general optical microscopy, which has limitations in EV observation due to the diffraction limit, and can quickly check the surface properties of EVs.

## 3. Nanobiomaterial-Based Exosome Biosensor

Several studies on the electrochemical, electrical, colorimetric, and fluorescence detection of exosomes have been reported. The developed sensor, based on electrochemistry and electricity, has the advantages of being small-sized, label-free [84], and having a fast detection time [85]. The colorimetric and fluorescent biosensor can visually confirm the presence of target molecules, making their combination a suitable analytical method for point-of-care diagnosis [86]. However, most sensors have low sensitivity compared to ELISA and RT-PCR technologies, which have proven their performance in disease diagnosis [87]. This section introduces nanomaterials to improve the sensitivity of next-generation point-of-care biosensors for exosome detection.

### 3.1. Novel Metal-Based Biosensor

Precious metal materials, such as gold, silver, platinum, and iridium are crucial in electrochemical and optical biosensors, owing to their excellent biocompatibility [88,89], electrochemical and electrical properties, and unique photoelectric properties [90,91]. The electrical properties of electrochemical sensors facilitate the functionalization of the metal surface, and the introduction of nanoparticles can increase the active area of the target and probe, leading to improved sensitivity of the sensor [92]. Regarding optical sensors, a sensor that utilizes the change in color intensity according to the structure of the metal nanoparticles is being developed [93]. In the field of fluorescence, metal nanoparticles are combined with a fluorescent probe to improve stability [94], with the size of metal nanoparticles enhancing binding to ligands [95]. This could improve the response speed of the sensor. The sensor applications involving these precious metal nanoparticles are focused on improving the sensitivity of the sensor; precious metal particles of the same material show different characteristics depending on the size and shape of the particles. This could be a good option for further improvement [96].

Wu et al. fabricated a gold nanoisland (AuNIs)-based localized surface plasmon resonance (LSPR) biosensor decorated with silver nanoparticles to detect glioblastoma-derived exosomes [97]. The proposed sensor (Figure 2A), composed of a biotinylated antibody targeting MCT4 on the exosome surface, has excellent corrosion resistance, ease of nanostructure fabrication, and the ability to generate plasmon resonance through light in the mid-range of visible light and in the visible light of Ag. Dynamic range and stability were increased by generating plasmon resonance in the blue range, which is at the end of the spectrum, and the antibody was fixed on the sensor surface via S-Ag binding of biotin and silver. Based on the binding of the antibody and exosome, a signal was generated by changing the refractive index of the surroundings. Based on the LSPR response versus exosome concentration (Figure 2B), the Au@AuNIs of the proposed sensor amplified the LSPR signal compared to normal AuNIs. In the mouse model, this sensor detected exosomes in the linear range of 7 × 10^2^~8.8 × 10^7^ particles/mL, and simultaneously showed a detection limit of 7 × 10^2^ particles/mL.

Su et al. reported a fluorescence-based method for exosome detection [49]. They proposed a three-dimensional DNA motor-based exosome analysis platform. The DNA motor consisted of gold nanoparticles bound to a fluorescently labeled substrate strand, and the motor strand was locked with an aptamer (Figure 2C). In response to the target molecule and aptamer, dehybridization of the aptamer bound to the motor strand promoted endonuclease activity in the motor strand. The activated endonuclease cleaved the fluorescently labeled substrate strand, thereby restoring the fluorescence [98]. This method had a detection limit of 8.2 × 10^3^ particles/mL in phosphate-buffered saline (PBS) and showed high selectivity (Figure 2D).

Geng et al. developed microfluidic chip-based LSPR-based plasmonic biosensors composed of gold nanoellipsoid arrays [50]. The proposed sensor was fabricated using an anodized aluminum oxide to fix the gold nanospheroids template, by an electron beam as a thin film, which enabled the inexpensive and mass production of metal nanostructures capable of detection in 50-μL samples. Sensor performance evaluation was performed with exosomes in the concentration range of 1.8 × 10^3^~1.8 × 10^7^ particles/mL, with a detection limit of 1.8 × 10^3^ particles/mL.

### 3.2. Transition Metal Chalcogenide-Based Biosensors

Two-dimensional (2D) transition metal dichalcogenides (TMDC), such as molybdenum disulfide (MoS_2_) and MoSe_2_, and nanocomposites, such as MXene, have an excellent surface-to-volume ratio, low bandgap energy, low cytotoxicity compared to carbon materials, and functional and catalytic properties of TMDC [99,100,101]. MXene is a nanomaterial used in electrical, electrochemical, and optical biosensors owing to its adjustable band structure, excellent electrical conductivity, and high surface chemical activity [102,103,104]. As the layered structure is formed by 2D TMDC and MXene van der Waals forces, formed layers can be separated, and single- or multiple-layer structures exhibit different conductivity and bandgap characteristics. Therefore, TMDC materials are widely used on the sensor surface because their unique properties improve the sensitivity of a sensor [105,106,107].

Dai et al. proposed an electrochemiluminescence (ECL) and photothermal dual-mode biosensor based on black phosphorus quantum dots (BPQD) and MXene [74]. Figure 3A shows the schematic of the proposed sensor. BPQDs have attracted attention in the field of ECL and are known to increase the ECL strength by acting as a catalyst for Ru(dcbpy)_3_^2+^ oxidation. BPQD, monomolecularly combined with Ru(dcbpy)_3_^2+^ through electrostatic interactions, shortens the electron transport distance and reduces energy loss, thereby increasing the efficiency of the sensor. As both MXene and BPQD that are used as support have photothermal properties, the sensor is now an ECL/photothermal biosensor that can detect exosomes in the linear range of 1.1 × 10^5^~1.1 × 10^10^ particles/mL, with a detection limit of 3.7 × 10^4^ particles/mL (Figure 3B).

Lee et al. proposed a DNA aptamer and a MoS_2_-based interdigitated micro-gap electrode (IDMGE) system for exosome detection (Figure 3C) [33]. IDMGE has a larger surface area than the electrode area and enhances the capacitance signal through the micro-gap electrode spacing. Application of the CD63 aptamer as a probe allows for lower production cost and faster synthesis while replacing the CD63 antibody. The proposed biosensor could detect exosomes in samples diluted with 100% human serum with a detection limit of 2.2 × 10^3^ particles/mL (Figure 3D).

### 3.3. Carbon-Based Biosensors

Carbon-based materials, such as fullerenes, carbon nanotubes (CNTs), graphene, and carbon dots, exhibit different properties depending on the arrangement of carbon. CNTs are one-dimensional carbon nanomaterials with excellent thermal and chemical stability, excellent conductivity, and a larger aspect ratio than ordinary materials [108,109]. Graphene is a representative 2D carbon nanomaterial with excellent thermal stability, optical properties, and conductivity, and its performance as a sensor can be improved based on the characteristics of the 2D materials that have a large specific surface area [110,111,112].

An electrical biosensor uses changes in current, voltage, impedance, and capacitance that occur because of a binding event between a working electrode composed of an antibody and an aptamer that recognizes a target biomolecule as detection signals [113]. This type of sensor does not require a separate label to detect a target and is used as a sensor for point-of-care diagnosis owing to its advantages of low power, easy miniaturization, and the requirement of a small amount of sample for analysis. Field-effect transistor (FET) measurement platforms that use transistors are representative. Recently, interdigitated electrode systems [114], large-area electrode systems, and new carbon materials are being actively studied to improve sensitivity. The application of one-dimensional materials, such as CNTs, conductive polymer nanowires, and silicon nanowires, is known to improve the sensitivity of FET sensors and increase the current switching characteristics [115,116]. In particular, the application of a modified carbon material improved the performance of the biosensor owing to its ability to control the insulation and conductivity of the material. Recently, studies have reported sensors to which 2D materials, such as graphene and graphene oxide, have been applied. These materials enable easy control of thickness, while improving their electrical properties [117].

Ye et al. proposed a membrane-based biosensor consisting of graphene hydroxide and a CD44 antibody [34]. Graphene hydroxide induces a strong current signal based on its large specific surface area and excellent conductivity, and forms holes on the surface according to the stacked structure of graphene. Subsequently, graphene was functionalized with an antibody against the CD44 protein, which is abundantly present on the surface of breast cancer-derived exosomes, to detect cancer cell-derived exosomes. The proposed sensor was able to detect exosomes in the linear range of 2.5 × 10^4^~1 × 10^9^ particles/mL, with a detection limit of 9 × 10^3^ particles/mL.

Tsang et al. developed a graphene back-gate field-effect transistor composed of 1-pyrene butyric acid N-hydroxysuccinimide ester (PBASE) and an anti-CD63 antibody [35] (Figure 4A). Graphene has excellent biocompatibility and can improve the identification of biomolecules by having carboxyl, hydroxyl, and epoxy groups. The PBASE linker bound to graphene immobilizes the antibody on the graphene surface and changes the dielectric constant by changing the thickness according to exosome binding (Figure 4B). Figure 4C shows the time-dependent sensor signal change according to the exosome concentration in PBS. The proposed sensor coupled with the CD63 tetraspanin of exosomes detected exosomes with a detection limit of 1.8 × 10^5^ particles/mL.

Zhang et al. proposed a field-effect transistor composed of reduced graphene oxide and an anti-CD63 antibody probe [36]. Due to its fast reaction and low detection limits, reduced graphene oxide has been used to detect DNA and tobramycin. The proposed sensor could detect exosomes diluted in HepG2 cell samples with a detection limit of 3.3 × 10^4^ particles/mL.

Ramadan et al. fabricated an exosome sensor with carbon dots (CD) and anti-CD63 antibodies and applied it to graphene field-effect transistors (Figure 4D) [37]. The application of nanostructures improves biomaterial accessibility on the sensor surface, and the spherical structure of CD promotes higher diffusion than planar structures. In addition, CD can complement the performance of graphene field-effect transistors owing to their low toxicity, high biocompatibility, and high surface-area-to-volume ratio. CD bound to graphene showed a greater Dirac voltage shift than under normal graphene conditions, as exosomes were bound, which is consistent with the negative charge effect of exosomes (Figure 4E,F). The manufactured biosensor can detect exosomes with a detection limit of 10^5^ particles/mL.

### 3.4. Metal-Organic Framework-Based Biosensors

MOFs are porous materials with large surface areas composed of organic linkers and metals or metal clusters, and have been utilized in drug delivery [118], chemical sensors [119], and catalysis [120] owing to their excellent chemical durability and flexibility in structural adjustment. MOFs have several advantages over metal nanoparticles and graphene and have been applied in biosensors. The MOF surface provides excellent binding to probes, such as antibodies and aptamers, based on a conjugated π-electron system [121]. Additionally, the application of flexible porosity and organic materials can increase the active area of the sensor and provide additional reactivity [122]. Finally, as a carrier, MOFs exhibit good binding properties between nucleic acids and substances, such as ions and enzymes, and can be encapsulated and stored. Therefore, the stability of the enzyme applied to the sensor can be improved, or the distribution of nanoparticles can be controlled. These organic and inorganic composites can improve the performance of sensors as nanobiohybrid materials [123].

Chen et al. reported an MOF structure-based SPR detection platform for exosome detection [51]. Figure 5A shows the schematic of the proposed sensor. Two-dimensional materials, which have recently been in the spotlight, have high specific surface areas, electron mobilities, and light absorptivities. Therefore, they are widely used to enhance SPR signals [124]. However, the resolution and quality factors tend to decrease even though the sensitivity of the sensor is improved by the interference of 2D materials [125]. The Cu-based MOF used by Chen et al. was synthesized using a hydrothermal method and showed high near-infrared absorption due to the d-d band transition of Cu^2+^ and its 2D ultrathin nature. In addition, the reduction in the resolution and quality factor of the SPR signal, which is a disadvantage of 2D materials, was improved. Specifically, the refractive-index sensitivity increased from 98°/RIU to 137°/RIU (Figure 5B). The detection system immobilizes the multifunctional peptide on a gold electrode modified with an MOF. Thereafter, the PD-L1 exosome detection ability was confirmed. The detection limit was 16.7 particles/mL, and also showed selectivity in human serum samples. The recovery rate of the sensor ranged from 93.43 to 102.35%.

Li et al. used a CD63 antibody, a zeolite imidazolate framework-8 encapsulated in glucose dehydrogenase (GDH), and a zirconium metal-organic framework loaded with a K_3_[Fe(CN)_6_] molecule to enhance electroactivity [38] (Figure 5C). A high-sensitivity self-powered biosensor was manufactured using the dual application. Attempts have been made to utilize the metal-organic framework as a catalyst in drug delivery owing to its large specific surface area and porosity. Recently, an investigation to maintain enzyme activity for a long period through encapsulation has been performed [126]. In this study, GDH@ZIF-8 was applied to the anode to improve the catalytic reaction and stability. To the cathode, a framework in which K_3_[Fe(CN)_6_] molecules were encapsulated was applied to induce voltage changes according to exosome detection. Owing to its high stability and sensitivity, the proposed biosensor showed linearity for the exosome concentration of 1.0 × 10^3^~1.0 × 10^8^ particles/mL and was able to detect exosomes with a detection limit of 3 × 10^2^ particles/mL (Figure 5D). Another study conducted by Li et al. proposed an electrochemical biosensor to cost-effectively detect exosomes based on MOF-applied paper, screen-printed electrodes, and aptamers [40]. UiO-66, composed of zirconium ions and organic linkers, was applied to paper connected with screen electrodes, such as Zr-MOFs, with high biocompatibility, and formed Zr-O-P bonds with phosphate heads, which are abundantly present within the exosome phospholipid bilayer. Aptamers recognize the CD63 protein in exosomes bound to Zr-MOF and trigger a hybridization chain reaction, and act as DNAzymes to generate an electrochemical signal. The proposed sensor can detect exosomes with a limit of 5 × 10^3^ particles/mL, with low cost and convenient operability.

## 4. Future Perspectives

Thus far, we have summarized various biosensors to detect exosomes. Table 1 summarizes the exosome sensor’s detection range and limits according to the nanoparticle, detection method, and probe type. The exosome biosensors developed till now have largely been electrochemical, electrical, colorimetric, and fluorescent. Electrochemical detection does not require a separate marker to detect exosomes, is cost-effective, and has the advantage of observing the live-state activity of cells and exosomes. However, it is difficult to specify the source. The electrical detection of exosomes also does not require a separate marker and is a useful detection method for on-site diagnosis owing to its low power and easy miniaturization of the sensor. However, the electrical signal response based on existing FETs and electrodes is simple, and disadvantages, such as difficulty in signal interpretation, high signal-to-noise ratio, and difficult measurement in fluid conditions, must be compensated for. Moreover, colorimetric and fluorescent biosensors require separate labels to detect exosomes, have poor sensitivity in colored samples, and have photobleaching potential for fluorescence signals, thereby limiting the lifetime of the sensor. Therefore, future exosome-detection biosensors should be developed considering these characteristics. In addition, sensors applied with aptamers as detection probes reported lower or similar detection limits compared to antibody-based sensors, proving that they can replace antibodies. Applying aptamer probes enables mass production at a lower price than antibodies and provides advantages in sensor reuse [127,128]. Based on existing research cases for detecting cancer cell-derived exosomes, exosomes can be utilized as a decisive factor for early cancer diagnosis through biopsy and can act as an essential factor in the field of cell analysis. Exosomes can be utilized as a decisive factor for the early diagnosis of cancer through biopsy and can also serve as an essential factor in the field of cell analysis. The surgical biopsy procedure used in the existing cancer diagnosis requires a long time and high cost. However, using a biosensor would enable cost reduction and early diagnosis of cancer. Therefore, the development of biosensors for the molecular diagnosis of cancer, based on exosomes, is crucial.

## Figures and Tables

**Figure 1 biosensors-12-00648-f001:**
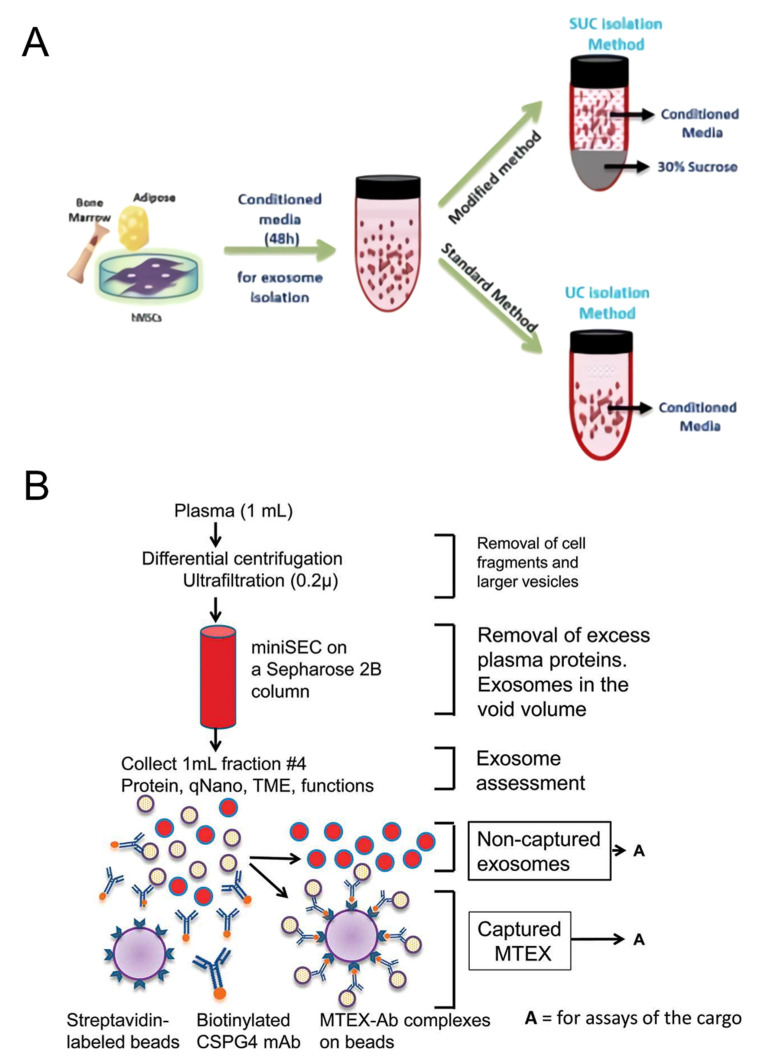
(**A**) Ultracentrifugation and sucrose cushion ultracentrifugation processes to isolate exosomes from stem cell cultures, (**B**) schematic diagram of an immunoaffinity-based separation process to precisely capture melanoma-derived exosomes. This was reproduced with permission from [80], published by Nature, 2018, and reprinted with permission from [81]. Copyright 2018, Taylor & Francis.

**Figure 2 biosensors-12-00648-f002:**
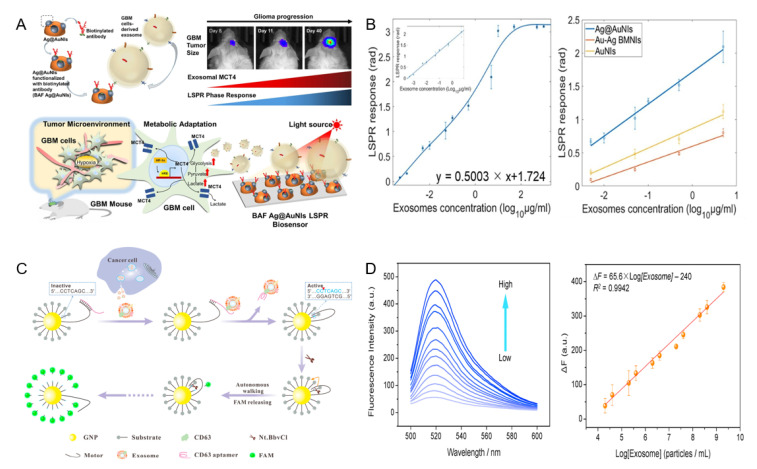
(**A**) Schematic diagram of gold nanoisland−based LSPR biosensor, (**B**) performance evaluation of gold nanoisland-based sensor according to exosome concentration versus LSPR response, (**C**) detection strategy of gold nanoparticle DNA motor-based fluorescent sensor, and (**D**) performance evaluation of fluorescent sensor according to wavelength and exosome concentration. This was reproduced with permission from [49,97], published by Elsevier, 2022 and 2020, respectively.

**Figure 3 biosensors-12-00648-f003:**
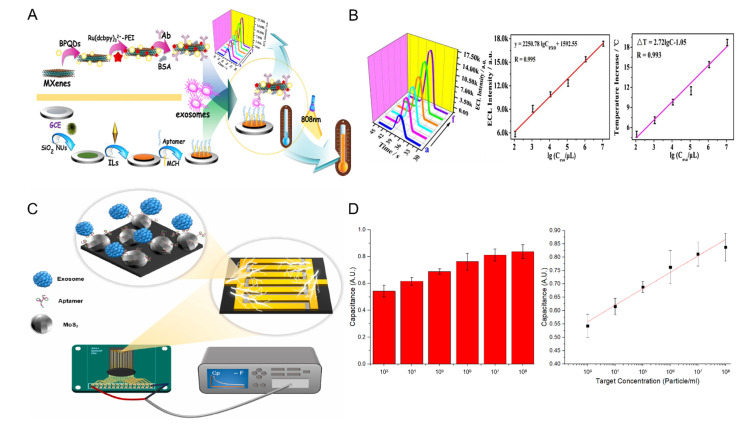
(**A**) Schematic diagram of ECL/photothermal dual biosensor composed of BPQD and MXene, (**B**) sensor performance evaluation under ECL and photothermal conditions, (**C**) schematic diagram of electric biosensor composed of MoS_2_ and IDMGE, and (**D**) evaluation of sensor detection performance through exosome concentration versus capacitance response. This was reproduced with permission from [33,74], published by Elsevier, 2020 and 2022, respectively.

**Figure 4 biosensors-12-00648-f004:**
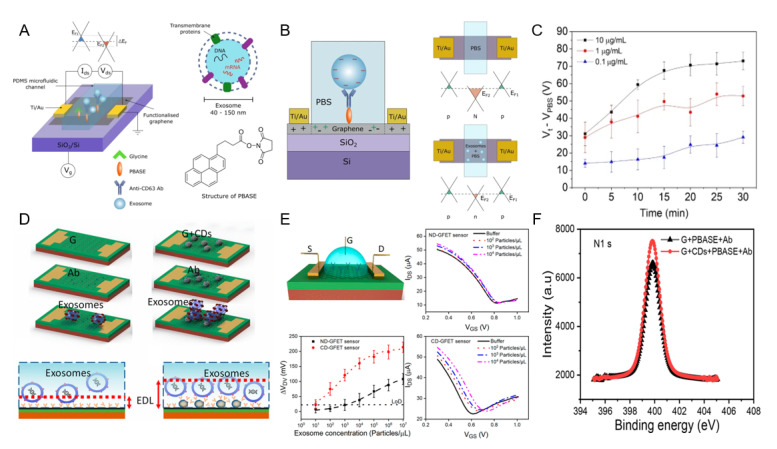
(**A**) Configuration of back-gate field effect transistor, (**B**) exosome detection principle of the proposed sensor, (**C**) sensor response according to exosome concentration in PBS buffer condition, (**D**) carbon dot applied graphene-based FET Schematic diagram of the sensor, (**E**) Comparison of exosome detection performance of sensors according to carbon dot application, and (**F**) sensor performance evaluation depending on whether carbon dot is applied. This was reproduced with permission from [35], published by Nature, 2019 and reprinted with permission from [37]. Copyright 2021 American Chemical Society.

**Figure 5 biosensors-12-00648-f005:**
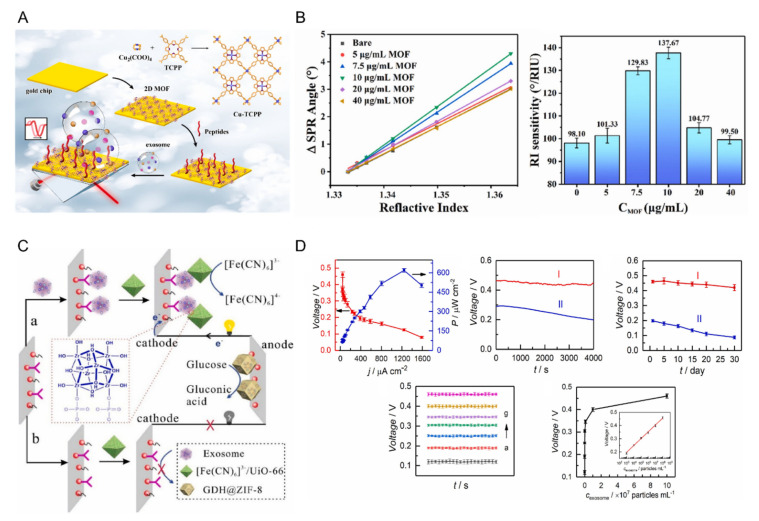
(**A**) Schematic diagram of Cu-based MOF-based SPR sensor, (**B**) SPR performance improvement according to MOF application, (**C**) Zr-based MOF-based self-powered sensor configuration, and (**D**) exosome detection performance of the proposed sensor evaluation. This was reproduced with permission from [38,51], published by Elsevier, 2022 and 2021, respectively.

**Table 1 biosensors-12-00648-t001:** Various nanomaterial-based exosome detection biosensors.

Materials	Nanoparticle	Detection Method	Probe	Detection Range	LOD	Ref
Nobel metal	Gold nanoisland	LSPR	Antibody	7 × 10^2^~8.8 × 10^7^ particles/mL	7 × 10^2^ particles/mL	[97]
	Gold nanoparticle	Fluorescence	Aptamer	2 × 10^4^~2 × 10^9^ particles/mL	8.2 × 10^3^ particles/mL	[49]
	Gold nanoellipsoid	LSPR	Antibody	1.8 × 10^3^~1.8 × 10^7^ particles/mL	1.8 × 10^3^ particles/mL	[50]
TMDC and MXene	MXene	Electrochemiluminescence	Antibody	1.1 × 10^5^~1.1 × 10^10^ particles/mL	3.7 × 10^4^ particles/mL	[74]
	MoS_2_	Electric	Aptamer	10^4^~10^8^ particles/mL	2.2 × 10^3^ particles/mL	[33]
Carbon	Hydroxylated Graphene	Electric	Antibody	2.5 × 10^4^~1 × 10^9^ particles/mL	9 × 10^3^ particles/mL	[34]
	Graphene	FET	Antibody	1.8 × 10^5^~1.8×10^7^ particles/mL	1.8×10^5^ particles/mL	[35]
	Reduced Graphene Oxide	FET	Antibody	3.3 × 10^4^~3.3 × 10^9^ particles/mL	3.3×10^4^ particles/mL	[36]
	Carbon Dot-Enhanced Graphene	FET	Antibody	10^5^~10^7^ particles/mL	10^5^ particles/mL	[37]
MOF	Cu based	SPR	Peptide	10^4^~5 × 10^6^ particles/mL	16.7 particles/mL	[51]
	Zr based	Electric	Antibody	10^3^~10^8^ particles/mL	3 × 10^2^ particles/mL	[38]
	Zr based	Electrochemical	Aptamer	1.7 × 10^4^~3.4 × 10^8^ particles/mL	5 × 10^3^ particles/mL	[40]

## Data Availability

Not applicable.

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
