# Peer review of "Introduction of Nanomaterials to Biosensors for Exosome Detection: Case Study for Cancer Analysis"

_biosensors, 2022, doi:10.3390/bios12080648_

Round 1
Reviewer 1 Report
After reading the manuscript these are my comments for the authors.
Page 2, line 48. Fix this: “direct detection and cancer cells circulating”
Page 2, line 49. Change “advantages: real-time” to ”advantages of real-time”
Page 2, line 65. “electricity [30-35], electrochemical[36-40], optical[41-45].” I do not think electricity is a platform, perhaps a “platform based on electricity”. Something like that. Fix this
Page 3, line 113. Change “advantages small size” to “advantages of small size”
Page 3, table 1. In here: “4â…¹10-4~50 μg/mL”, use the times symbol (×) not an x. Same here: “5â…¹103 particles/mL”
Page 3, table 1. Change “Electro chemical” to “Electrochemical”
Page 3, table 1. Table 1 is not referenced (called) in the manuscript. Describe this table in the text and comment further on why antibodies or aptamers are used as ligands.
Page 4, line 133-134. You establish “with the size of metal nanoparticles having a small effect[67]”, and then on line 137 you mention “depending on the size and shape of the particles”. It is confusing. There is dependence on size or not?
Page 4, line 145. Change “which is the end of the” to “which is at the end of the”
Page 5, line 156-157. Fix this: “Performance evaluation of fluorescent sensor according. Reproduced”
Page 5, line 185. I am not familiar with this term: “cambium”. Is it correct?
Page 6, figure 2. Figure 2D is of poor quality. Please replace it.
Page 7, line 242. Change “antibody of CD44 protein” to “antibody against the CD44 protein”
Page 7, line 256-257. “DNA detection and tobramycin because”, is it “DNA and tobramycin detection because”?
Page 7, line 266. Change “than that under” to “than under”
Page 8, figure 3. Rewrite the caption for figure 3D (Carbon point applied graphene-based FET Schematic diagram of the sensor) for clarity purposes. Use carbon dots as in the text, not carbon point. Same for the caption of figure 3F.
Page 8, line 296-297. “Therefore, they are widely used to enhance SPR signals [refer].” Reference?
Page 8, line 298. “2D materials, [].” Reference?
Page 9, figure 4. Better quality images should be used.
Page 9, line 305. “ability was confirmed (Fig. xxx c)”, ??
Page 9, line 320. “encapsulation has been performed [refer]”, reference?
Page 10, line 332-33. Remove commas from: “Zr-MOF, trigger a hybridization chain reaction, and act as DNAzymes”
Page 10, line 333-334. “The proposed sensor can detect exosomes with a detection limit of 5×103 particles/mL owing to its low cost and convenient operability”. I do not think the low cost and operability affect the detection limit, rewrite this sentence.
Page 10, section 3. Section 3 (Exosome Biosensors Without Nanomaterials) does not make sense in the light of the title of the review. Either change the title, erase this section, or use some of it as reference in the introduction.
Page 11, line 401. Change “potential for fluorescence” to “potential of fluorescence”
Page 11, section 4. The Future perspectives section comes short, here we expect to see the expertise of the writers on the topic of the review. Besides commenting advantages and disadvantages the authors should criticized further, are any of the reports in the literature trying to detect exosomes in real samples? Particles/ml makes sense to me, µg/mL does not. A word on this? ELISA and PCR take time, despite this, are they good enough to be used in liquid biopsies? Waiting a day for a result makes sense to me as cancers take years to develop. A word on this? Disposable sensors instead of regenerable sensors. A word on this? Antibody stability is always an issue, especially when pursuing regeneration. A word on this?
Page 12, line 426-435. I guess this part should be erased: “References must be numbered in order of appearance in the text (including citations in tables and legends) and listed individ- 426 ually at the end of the manuscript. We recommend preparing the references with a bibliography software package, such as 427 EndNote, ReferenceManager or Zotero to avoid typing mistakes and duplicated references. Include the digital object identifier 428 (DOI) for all references where available. 429 430 Citations and references in the Supplementary Materials are permitted provided that they also appear in the reference list here. 431 432 In the text, reference numbers should be placed in square brackets [ ] and placed before the punctuation; for example [1], [1–3] 433 or [1,3]. For embedded citations in the text with pagination, use both parentheses and brackets to indicate the reference number 434 and page numbers; for example [5] (p. 10), or [6] (pp. 101–105).”
Author Response
The authors agree with the reviewers' comments. The paper has been edited based on your comments. Please check the attached file for detailed edits.
Thank you for your valuable comments.

Reviewer 2 Report
The manuscript of Myoungro Lee et al. is an review concerning the recent methods of detection of extracellular vesicles for cancer diagnostics purposes. It contains a description of several nano-biomaterial-based biosensors for exosomes detection including novel metal-based biosensors, transition metal chalcogenide-based biosensors, carbon-based biosensors, metal-organic framework-based biosensors, as well as chosen examples of exosome biosensors without nanomaterials. The last section concerns the future perspectives in exosome detection. The manuscript is well written, but some editing errors occur. The references used are relevant to the research, but the authors should extend scope of the references.
The manuscript is within the scope of Biosensors Journal, but before acceptance extensive corrections are necessary and some explanations should be provided. Below are my detailed comments about the manuscript.
Major comments:
1) The manuscript covers the scopes of some other already published reviews concerning this matter, for example:
https://doi.org/10.2147/IJN.S333969,
https://doi.org/10.3390/ijms23020868,
https://doi.org/10.1007/s00216-020-03000-0,
https://doi.org/10.1016/j.aca.2020.02.041,
https://doi.org/10.1016/j.omtn.2022.04.011
Therefore, it is difficult to grasp what the novelty of this review is about and what is its purpose.
In my opinion, the authors should seriously consider broadening the scope of this review if it is to be accepted for publication. They should substantially modify and revise their manuscript by extending its scope and including some additional, new insights into biosensors for detection of EVs and their problems.
2) In the introduction section, the authors are focusing on the detection techniques e.g. “Recently, efforts have been made to detect exosomes based on various detection 64 platforms such as electricity [30-35], electrochemical[36-40], optical[41-45].” However, it should be noted that the use of the extracellular vesicles (EVs) for diagnostic purposes require not only the step of detection but also the necessary step of EV classification. Different EVs exhibit different biogenesis, molecular, optical, chemical, biological, and morphometric parameters such as size, density, and shape. The authors should indicate that the practical use of EVs in diagnostic will be based not only on the detection methods but also on characterization of their properties: morphological, chemical, optical signatures/fingerprints. They are not presenting recent methods focused on characterization of the EV properties for diagnostics purposes. The author should consider updating the Introduction Section based on the recent advances in EV phenotyping. Below are some examples to consider:
https://doi.org/10.1080/20013078.2020.1790158
https://doi.org/10.1016/j.cytogfr.2019.12.007
https://doi.org/10.3390/ijms21186543
https://doi.org/10.1080/20013078.2019.1710020
https://doi.org/10.1016/j.ajpath.2021.08.005
https://doi.org/10.3390/ijms21228723
The phenotypic examination of a single EV or subpopulations of EVs is a new technology based on the optical techniques or their combination with immunology. Phenotyping technology the oncologists more complex analysis of tumor-related exosomes to better conduct tumor pathophysiological research, early detection, diagnosis, and clinical management. In my opinion, the authors should include an overview of the recent methods of visible and quantitative characterization/ phenotyping of EVs, because the diagnostic potential of the EVs is no only based on their detection, but is more related to the their classification and differentiation based on their properties. This issue is very barely discussed in other reviews already published.
3) Moreover, it should be pointed out in this review for which kind of EVs’ samples these methods of EVs detection are dedicated. It is possible to use them to detect the EVs’ extracted from tissue, precipitated from human blood plasma, urine, or cell cultures? Different types of EVs’ extraction/collection require the use of specific sample preparation procedures as sample purification (e.g. ultracentrifugation) and isolation of the EV, which can affect EVs’ properties as their cargo density. In my opinion, the authors should indicate this aspects and indicate in which way the EVs’ samples should be prepared for a specific biosensing method. Several years ago, it was indicated that for the use of EVs as biomarkers of disease the standardization of collection, isolation and analysis is necessary (see https://doi.org/10.3402/jev.v2i0.20360). Unfortunately, this issue is not considered in this review, but in my opinion it should be included, because these factors have a crucial impact on the possibility of using EVs for oncological diagnostic purposes at all.
4) In my opinion, in the section “Exosome Biosensors Without Nanomaterials”, the authors did not included some interesting optical biosensor or EVs sensing method. They should consider description some of them in this section.
https://doi.org/10.1038/srep37246
https://doi.org/10.1016/j.ajpath.2021.08.005
doi:10.7150/thno.33683
https://doi.org/10.1021/acs.analchem.8b01831
https://doi.org/10.1373/clinchem.2018.291963
https://www.sciencedirect.com/science/article/abs/pii/S0925400521014611
Minor comments:
1) There is no reference to Table 1 in the text.
2) Figure and its caption should be on the same page.
3) It will be valuable to standardize the unit of the dynamic range in Table 1, because now the are expressed as ug/mL or particles/mL.
4) Explain all abbreviation used in the manuscript before using them.
5) Line 305: “(Fig. xxx c)”?
6)Line 13: The authors with equal contribution were not indicated.
7) A lot of missing spaces see line 50, 52, 73, 100, 181, 184, 188, 221, 224 and a lot of others.
Author Response

(The authors gave the same response as above.)

Round 2
Reviewer 2 Report
I appreciate some of the improvements made to the manuscript (particularly the introducing the section entitled "Pretreatment of cell-derived exosomes". unification of units etc.), but in relation to the main points raised in my previous review, these are insufficient in my view.
As originally written in my previous review report, the authors should seriously consider broadening the scope of this review if it is to be accepted for publication. In the literature there are several similar reviews already published (as indicated previously), which cover the same scope as the submitted manuscript. Therefore, I really do not understand the need to publish a review paper entitled ‘INTRODUCTION of the Nanmaterials to Biosensors (...)’, if there are already published reviews covering the same subject matter. In my opinion, the authors did not include significant modifications to the manuscript.
In their response, the authors generally agree with my previous suggestions, but the changes made do not make this apparent and, in my view, these corrections are too cursory.
1) I suggested including some additional, new insights into biosensors for detection of EVs and their problems, and this was made partially (e.g., introducing the section entitled Pretreatment of cell-derived exosomes).
2) In order to use exosomes for cancer diagnosis (as indicated in the title and in the introduction) not only is the step of detection of exosomes necessary, but I my opinion rather their phenotyping (indicating changes in their pathogenesis-related properties as cargo content, size distribution, density etc.) is more crucial. Also, with this comments authors agreed in their response, but the issue of exosomes phenotyping was neglected in their review at all. The authors focused only on biomaterials-based biosensors, even if there are many similar works in the same scope. There are many reported attempts to use different measurement/ imaging techniques for exosomes phenotyping:
-IR spectroscopy(ATR-FTIR, Raman spectroscopy) e.g.:
doi.org/10.1371/journal.pone.0235214
doi.org/10.1016/j.ejps.2022.106135
-Optical methods (fluorescence, quantitative phase imaging, dSTORM, etc.) e.g.:
https://doi.org/10.1038/s41598-019-56375-1
https://doi.org/10.1016/j.ajpath.2021.08.005
https://doi.org/10.1002/jev2.12191
-Cryo-transmission electron microscopy, e.g.:
https://doi.org/10.1111/jth.12554
and many others (including these indicated in the previous review report). But the authors decided not to extend the scope of the review, which is wrong in my opinion.
3) I suggested that the authors should extend the scope of the introduction by indicating the recent development in the phenotyping of extracelullar vesicles, but the authors barely included this suggestion. In the previously indicated sentence: ‘Recently, efforts have been made to detect exosomes based on various detection platforms such as electricity [30-35], electrochemical [36-40], optical [41-45].’ They did not include any additional references.
4) In the manuscript are still the editing errors, which should be corrected after resubmission e.g.:
-The quality and resolution of the figures are very poor (Fig.1A, Fig.3, Fig.5D)
-Still there are a lot of format errors including lack of spaces, e.g. lines 49-52, unnecessary dots, e.g. line 417, wrong type of font, e.g. line 7
-The authors did not indicate the authors with equal contribution even if this is mentioned in line 13.
-please decide if you wrote “Republic of Korea;” or “Korea” in the affiliations.
- I don’t understand the expression in line 67: “(..) for diagnosing small sEV and LEV diseases”: small sEV?, LEV?
Author Response

(The authors gave the same response as above.)

Round 3
Reviewer 2 Report
I would like to thank the authors for taking into account my comments and the changes made, which will distinguish this review paper from others on the same topic. These additional sections significantly improved the scientific level of this work.